# Spatio-Temporal Regulation of Notch Activation in Asymmetrically Dividing Sensory Organ Precursor Cells in *Drosophila melanogaster* Epithelium

**DOI:** 10.3390/cells13131133

**Published:** 2024-06-30

**Authors:** Mathieu Pinot, Roland Le Borgne

**Affiliations:** Univ Rennes, Centre National de la Recherche Scientifique UMR 6290, IGDR (Institut de Génétique et Développement de Rennes), F-35000 Rennes, France

**Keywords:** Notch signaling, asymmetric cell division, cell fate determinants, cell polarity, epithelial cells, *Drosophila melanogaster*

## Abstract

The Notch communication pathway, discovered in *Drosophila* over 100 years ago, regulates a wide range of intra-lineage decisions in metazoans. The division of the *Drosophila* mechanosensory organ precursor is the archetype of asymmetric cell division in which differential Notch activation takes place at cytokinesis. Here, we review the molecular mechanisms by which epithelial cell polarity, cell cycle and intracellular trafficking participate in controlling the directionality, subcellular localization and temporality of mechanosensitive Notch receptor activation in cytokinesis.

## 1. Introduction

Notch is the receptor of an evolutionarily conserved cell–cell signaling pathway. In both vertebrates and invertebrates, it is regulating a large number of developmental processes, whose deregulation is at the root of many developmental disorders, congenital disorders as well as tumoral and non-tumoral pathologies (for reviews see reviews [1,2,3,4,5,6]). At the cellular and tissue level, Notch plays a central role in cell fate acquisition. 

A very well-characterized example of Notch-dependent fate decisions comes from studies carried out on mechanosensory organs (SOs) present on the dorsal thorax of *Drosophila* pupae, called pupal notum [7,8,9]. The notum is a single-layer epithelium composed of epidermal cells and SOs. Epidermal cells divide symmetrically within the plane of the epithelium to give rise to two identical daughter cells, a process allowing an increase in epithelia surface. The cell divisions giving rise to SOs are governed by complex mechanisms. Each SO is produced by a single precursor, the sensory organ precursor (SOP), that is selected within a group of equipotent cells, the proneural cluster [10]. This selection occurs thanks to Notch-mediated lateral inhibition [10]. The SOP divides within the plane of the epithelium and along the antero-posterior axis of the pupae to generate a larger posterior cell and a smaller anterior cell [7,8,11]. The identity of daughter cells is reliant on the unequal segregation of two cell fate determinants in the anterior cell, Numb and Neur [12,13]. Neuralized (Neur) is an E3 ubiquitin ligase that positively regulates the activity of the Notch ligands by endocytosis [13,14,15,16]. Numb is a membrane-associated phosphotyrosine protein that inhibits Notch signaling by regulating endosomal sorting of Notch [17,18,19]. As a consequence of unequal partitioning of Neur and Numb, Notch is activated in the posterior cell that adopts the pIIa fate, while its daughter cell adopts the pIIb fate. The pIIb cell is the precursor of the two internal cells, whereas pIIa will give rise to the external cells of sensory organs, respectively [7,8]. Thanks to sophisticated genetic toolboxes including unbiased genetic screens and genome manipulation/genome editing, coupled with cell biology approaches which include non-invasive and quantitative live-imaging, the apparently simple model of the SOP has made it possible to lay the foundation for a number of fundamental processes required for directionality in Notch-dependent fate acquisition. Deciphering this mechanism in *D. melanogaster* also became an archetype for the understanding of the molecular mechanism governing the establishment of cell identity following the asymmetric cell division of progenitor cells in other species and tissues [20,21].

The current knowledge addressing the mechanisms leading to the selection of SOP and the regulation of the antero-posterior orientation of SOP division have been nicely presented in excellent recent reviews [9,22] and will not be covered in this review article. Here, we will first introduce the basic principles of Notch signaling and asymmetric cell division (ACD), to then review and discuss the current knowledge addressing the following questions, 1. how asymmetry of cell fate determinants is established at SOP mitosis and biases signaling at cytokinesis, 2. how epithelial cell–cell contacts are remodeled throughout SOP division to maintain tissue integrity while assembling a signaling interface between daughter cells, and 3. how polarized trafficking and endocytosis of Notch signaling components are regulated at the signaling interface, and all this to illustrate how these events are coordinated in time and space to ensure fidelity and robustness in the pIIa/pIIb binary fate decision. 

## 2. Discovery and Overview of the Notch Signaling Pathway

The origins of the Notch pathway lie in the pioneering work of T.H. Morgan and D.F. Poulson in *Drosophila*, with the identification of small deficiencies on the X chromosome, which in heterozygous females were giving rise to serrations on the wing, hence the name *Notch* [23,24]. Notch is characterized as a dominant haplo-insufficient mutation. Embryonic lethality is caused by serious developmental disturbances, among which is hypertrophy and disorganization of the nervous system (ref. [23]; for review, see [1]). The latter, named the neurogenic phenotype, is a consequence of a switch of fate of epidermal precursors into neuroblasts that continue their normal differentiation, causing the hypertrophy of the nervous system [25]. Unbiased genetic approaches then further demonstrated the powerfulness of the *Drosophila model*, as illustrated by the screening of embryonic phenotypes that led to the identification of a group of six loci exhibiting neurogenic phenotypes reminiscent of that seen in *Notch* mutant embryos [26,27]. These loci included *Delta*, encoding a type I transmembrane protein, one of the Notch ligands, *Enhancer of Split* (*E*(*Spl*)), encoding a helix-loop-helix Notch target, the transcriptional co-activator *mastermind* (*mam*), *big brain* (*bib*), encoding a nonselective cation channel of the aquaporin family, and the E3 ligase *neuralized* (*neur*), which are directly implicated in the core Notch signaling pathway (Figure 1). Within the same decade, the *Notch* coding gene was isolated and sequenced revealing Notch to be a large cell surface, single-pass type I glycoprotein containing epidermal growth factor (EGF)-like repeats in its extracellular domain [28,29,30]. In the late 1980s, Notch orthologs were identified in *Caenorhabditis elegans*, which, as in *Drosophila*, have been shown to be essential for cell interactions and cell fate specifications during worm development [31,32,33,34,35]. Vertebrate *Notch* receptors were identified and cloned shortly after [36,37], demonstrating that Notch is a conserved communication pathway in metazoans. Advances in gene cloning, unbiased genetic screens in *Drosophila* and *C. elegans* coupled with biochemical and cell biology approaches in vertebrate cells have enabled the Notch signaling pathway to be characterized at both cellular and molecular levels. 

Thus, the work carried out by a scientific community of researchers has led to the characterization of the Notch-dependent pleiotropic cell–cell communication pathway schematized in Figure 1. It is striking to note that the main components of the Notch pathway have been conserved over the course of evolution (Table 1). Notch signaling is essentially paracrine as its ligands of the DSL (for Delta and Serrate from *Drosophila* and Lag2 from *C. elegans*) family are also transmembrane proteins. The Notch receptor is activated by DSL ligands present at the surface of adjacent cells, referred to as signal-sending cells. In the receiving cell, signal activation involves three successive cleavages (for review see [38,39]). Notch is first processed in the *trans*-Golgi network by the Furin endoprotease to generate a functional heterodimeric receptor (S1 cleavage) [40]. In *Drosophila,* unlike in mammals, Notch at the plasma membrane is full-length and the S1 cleavage is dispensable for its activity [41]. A second ligand-induced cleavage of Notch at the extracellular S2 site by the ADAM/TACE Kuzbanian generates a membrane-bound activated form [42,43] which then undergoes an intramembranous cleavage at the S3 site by the g-secretase complex [44,45,46]. This last cleavage leads to the releasing of the active Notch intracellular domain (NICD) that is prone to be translocated to the nucleus, where it interacts with a DNA-binding protein called CSL (for human, CBF1; *Drosophila*, Suppressor of Hairless; *C. elegans*, Lag-1) to regulate the transcription of target genes [47,48] (Figure 1, Table 1). 

This mode of cell–cell paracrine communication is used repeatedly throughout development, especially in epithelial tissues [1,2]. The sequence of events leading to Notch activation requires key regulators combined with coordinated intracellular trafficking, so that the ligand can interact in trans with the receptor in the right place and at the right time. As we will describe in the following paragraphs, this spatio-temporal regulation has been particularly well characterized in SOPs, progenitors which divide asymmetrically to generate two distinct daughter cells, whose identity is solely governed by Delta/Notch communication. 

## 3. Establishment of the Asymmetry of Cell Fate Determinants 

### 3.1. From Polarization to Asymmetric Division of Sensory Organ Precursor Cells

SOP division is a well-studied model of asymmetric cell division (ACD), a process by which cell diversity is generated throughout development [7,9]. During ACD, a mother cell divides to generate two daughter cells with distinct identities and developmental potential. Asymmetry often relies on the unequal distribution of cell fate determinants, and its fundamental principles were to a large extent established in the models of the one-cell *Caenorhabditis elegans* embryo and *Drosophila* neuroblasts [49,50,51,52]. ACD can be subdivided into three steps: (1) the mother cell polarizes before entering mitosis; (2) cortical polarization dictates the asymmetric localization of cell fate determinants at one pole of the mother cell in the form of a calotte, a.k.a. as a crescent because of its characteristic cross-sectional shape; and then (3) the mitotic spindle aligns with the middle of the crescent of cell fate determinants, leading to their unequal segregation in only one of the daughter cells at cytokinesis. Consequently, cell fate determinants regulate the acquisition of cell fate in each of the daughter cells by their presence or absence. 

### 3.2. Polarization Drives Asymmetric Localization of Cell Fate Determinants

While the SOP in the pupal notum follows these basic rules as it asymmetrically divides, leading to the unequal partitioning of the two cell fate determinants Numb and Neur, there is an additional layer of complexity, related to the fact that SOPs are epithelial cells (Figure 2). SOPs are surrounded by and tightly bound to neighboring epidermal cells by cell–cell junctions, the apical adherens junctions (AJs) and the lateral septate junctions (SJs), which act as mechanical and paracellular diffusion barriers, respectively [9,53,54,55] (Figure 2B). Thus, SOPs are subjected to two types of polarization: (1) at the tissue scale, where planar cell polarity (PCP) mediated by Frizzled signaling orients SOP division along the antero-posterior axis [11,56]; (2) at the cellular level, where apical–basal polarity establishment and maintenance depends on the activity of cortical polarity proteins. Apical to AJ, the PAR complex is localized, composed of Bazooka (Baz, the *Drosophila* Par3)–DmPar6 (Par6)–atypical PKC (aPKC) complex, and localized at the SJ are Discs-large (Dlg) and Scribble (Scrib), which cooperate with Lethal (2) giant larvae (Lgl) [57,58]. During SOP mitosis, the asymmetric localization of Numb and Neuralized requires the remodeling of the polarity modules with the Baz-Par6-aPKC complex localizing at the posterior apico-lateral cortex, while, on a mutually exclusive basis, Dlg together with Partner of Inscuteable (Pins) accumulate at the anterior lateral cortex [54] (Figure 2C,C′). 

The assembly of the Baz-Par6-aPKC complex was shown to be regulated by Aurora A-dependent phosphorylation, leading to the proposal that the complex forms and localizes at the posterior cortex upon the entry into mitosis [59]. However, this view was revisited using real-time analyses, revealing that the Baz-Par6-aPKC complex occurs independently of AurA before entry into mitosis, and its location at the posterior pole is regulated by PCP [60]. It is interesting to note that while in interphase, the Par complex is restricted apically above the AJs, while upon entry into prophase, it extends laterally at the level of the AJs and SJs, suggesting a change in the cortical polarity at mitosis entry [54,60]. During mitosis, the Par complex exerts at least two functions. The first one is the aPKC-dependent phosphorylation of Numb, preventing it from localizing at the posterior cortex [61]. As a result, Numb is exclusively localized at the anterior cortex. It remains to be determined whether Neur-dependent phosphorylation of aPKC explains its unequal distribution. The second function of the Par complex, also mediated by aPKC-dependent phosphorylation, consists in restricting the function of the tumor-suppressor gene Lgl, required for Neur and Numb asymmetric localization, to the anterior cortex [62]. 

## 4. Remodeling of Cell–Cell Contacts during SOP Cytokinesis

### 4.1. Alignment of the Mitotic Spindle with Cortical Polarity

At SOP prometaphase, the mutual exclusion of Pins/Dlg and the Baz-Par6-aPKC module sets the asymmetric localization of Numb and Neur [54]. Then, the SOP divides along the a-p axis, a process relying on PCP to allow unequal partitioning of fate determinants [11,56]. This implies a coupling between mitotic spindle orientation and cortical polarity determinants to drive the unequal partitioning of Numb and Neur in the pIIb cell. At the anterior cortex, a complex composed of Pins, Mud and Dynein exerts pulling forces on astral microtubules [63,64,65]. Phosphorylation of Pins by aPKC and AurA is proposed to regulate the localization of this complex [66,67,68]. In symmetrically dividing epidermal cells, apical aPKC excludes Pins from the apical cortex, an activity needed to orient the spindle within the plane of the epithelium [67]. Based on these observations, in dividing SOP, by phosphorylating Pins at the posterior cortex, aPKC is proposed to induce Pins posterior exclusion. On the other hand, AurA, by phosphorylating Pins, may promote the localization of Pins-Dlg in the anterior cortex. In addition to these anterior cortical processes, the posteriorly localized Fz receptor and Dishevelled (Dsh), two core components of PCP signaling, recruit the Mud-Dynein complex to stabilize astral spindle microtubules in the posterior apical cortex [65,69]. This appears to be a cell cycle-regulated process as Cyclin A, the major cyclin driving the transition to the M-phase of the cell cycle, is recruited at the posterior cortex by Fz-Dsh in prophase. Cyclin A is in turn required for Mud-Dynein recruitment at the posterior apical cortex [70]. Thus, Pins and Fz-Dsh inputs in the anterior and posterior cortex, respectively, are responsible for recruiting the Mud-Dynein complex to ensure spindle orientation along the a-p axis and to couple the spindle orientation to the asymmetric localization of fate determinants. As the cortical domain of Pins-Dlg is located basally relative to the apical Fz-Dsh complex, the mitotic spindle is not parallel to the plane of the epithelium, as is the case in epidermal cells, but it is tilted towards the antero-basal pole [54]. In cytokinesis, because the cleavage furrow is positioned orthogonally to the mitotic spindle, one of the consequences of this spindle tilt is the formation of an anterior cell with a smaller apical pole compared to its posterior sister cell. In addition, the Lgl-dependent posterior enrichment of Rac and Cdc42 promotes a higher level of branched actin in the posterior cortex of the SOP, contributing to produce two daughter cells with distinct volumes [71].

### 4.2. Activation of Notch Takes Place during SOP Cytokinesis 

The generation of fully functional versions of Notch whose cytoplasmic domain was fused to GFP (NiGFP) was instrumental in determining the dynamics of Notch localization and signaling throughout SOP mitosis [72,73]. This elegant study from F. Schweisguth’s laboratory first revealed that low levels of NiGFP are detected in dividing SOPs and at the pIIa-pIIb interface compared with those detected between two epidermal cells. Second, NiGFP is best detected at the apical pIIa-pIIb interface, but is also present along the lateral interface 10 min after the onset of anaphase. Third, following the ligand-mediated proteolytic cleavage by g-secretase, the processed intracellular domain of Notch (NICD) fused to GFP is detected in the nucleus. Time-lapse imaging indicates that there is a progressive increase in NiGFP in pIIa nuclei, reaching a plateau 30 min after the onset of anaphase [72]. Nuclear NiGFP levels correlate with the Notch transcriptional response detected using a functional GFP-tagged *E*(*spl*)*M8-HLH* reporter [73]. Nuclear NiGFP is not detected in the pIIb cell and the ratio of nuclear NiGFP in pIIa to pIIb cells increased over time. Hence, the functional NiGFP allowed Notch activation to be monitored in vivo to determine that the directionality of Notch activation is established shortly after anaphase [72,73]. 

This study raised the fundamental question of the site of Notch activation, particularly as SOPs and their daughters express low levels of Delta and Notch compared with epidermal cells. Based on the SOP-specific expression of Neur and the unequal partitionning of Neur in the pIIb cell, it is generally viewed that pIIb is the main source of ligand-activating Notch in the pIIa cell [74], reducing the system to a private cell–cell communication between pIIb and pIIa, surrounded by epidermal cells that are ‘passive’ in terms of pIIa-pIIb fate acquisition. Among other issues, this raises the question of the polarized and coordinated transport of Delta and Notch to the pIIa-pIIb interface in cytokinesis, and their ability to interact in trans. Indeed, based on the fact that Delta and Notch are expressed at low levels and that their time residency at the plasma membrane is short, the possibility that a Delta from the pIIb cell might encounter a Notch on the surface of the pIIa cell seems akin to looking for a needle in a haystack. Membrane trafficking regulators are part of the answer (see also the paragraph on the additional contribution of membrane traffic in binary cell fate acquisition). For example, analyses of the borders of clones of cells expressing Notch-GFP indicate that Notch is selectively targeted towards the pIIa-pIIb interface at cytokinesis [55]. The clathrin adaptor AP-1 complex and stratum, a binding partner of Rab8, act upon the exit of the Golgi apparatus to regulate the transport of Notch and Sanpodo (Spdo)—a Notch interactor specifically expressed in SOPs and daughter cells (see below)—towards the basolateral plasma membrane [75,76,77]. However, while polarized trafficking of course plays an important role, as we shall see in the following paragraphs, the remodeling of polarity modules in the dividing SOP, unequal partitioning of cell fate determinants and the activity of Spdo play a major role in the directionality, timing and subcellular site of Notch activation.

### 4.3. Assembly of the Notch Signalling Interface at SOP Cytokinesis 

One of the remarkable features of proliferative epithelia, such as the pupal notum, is that the SOPs and epidermal cells divide without the tissue tearing. Analyses of the dynamics of the main components of the AJ and SJ showed that, when epidermal and SOP cells divide, the junctional complexes are remodeled without any loss of integrity of the epithelial barrier functions [53,78,79]. One of the characteristics of epithelial divisions is the formation of a characteristic long contact interface between daughter cells. The junction remodeling is particularly important at the cleavage furrow, where cell junctions with neighboring cells are disassembled to enable the subsequent assembly of cell–cell junctions between daughter cells [53,79]. While the remodeling of AJ and SJ during SOP cytokinesis is similar to that of epidermal cells, the asymmetric distribution of the Par complex and Pins-Dlg modules at mitosis results in an atypical polarity of the nascent pIIa-pIIb interface [55]. During cytokinesis, following AurA degradation, the Baz-Par6-aPKC complex is disassembled. Numb and Neuralized are released from the anterior cortex to regulate the activity of Notch and its ligands [19]. On their side, aPKC and Par6 are localized apically above the AJ, while Baz is relocalized at the SOP daughter cell interface in the form of apical and lateral clusters [55] (Figure 2D,D′). The formation of Baz apical and lateral clusters is unique to the SOP daughter interface and not observed in epidermal cells [55]. Interestingly, clonal analyses using Notch-GFP revealed that Notch is preferentially targeted to the pIIa-pIIb interface, where it localizes in dotted structures [55,72]. Interestingly, these structures colocalize with Baz. Baz appears to assemble clusters independently of Notch, while Notch clustering requires Baz [55]. Baz activity, in addition to being necessary for cluster assembly, contributes to the activation of Notch signaling. Spdo is also located in the Baz-Notch clusters and Spdo activity is required for their formation. Numb, which inhibits Notch, is not detected in Baz-Notch clusters. In contrast, Neuralized, which activates Delta signaling, is localized in these clusters, and Delta and Neur activities regulate the assembly and stability of Baz-Spdo-Notch clusters [55] (Figure 2E). 

Collectively, these data raise the possibility that Baz-Spdo-Notch-Neur clusters correspond to the sites where Delta can activate Notch in trans at cytokinesis. 

## 5. Role of Polarized Trafficking and Endocytosis in Notch Activation

### 5.1. Site of Production of the Notch Intracellular Domain (NICD)

As we have just seen, during cytokinesis, the two pools of Notch are at the pIIa-pIIb interface, one apical at the level of the AJs and the other located more basally under the midbody (Figure 2E). While both pools of Notch are positive for Baz-Spdo and Neur and transient [55], their contribution to Notch signaling appears to differ. Selective photobleaching (of NiGFP) or photoconversion (of NimMaple3) of the apical versus basal pool of Notch revealed that nuclear Notch primarily originates from the basally located Notch receptor [73]. While the basal pool of Notch is the major contributor to Notch activation, it appears that the apical pool also participates in NICD production [55,73,76]. However, it remains to be determined whether NICD is produced directly from the apical pIIa-pIIb interface or whether basolateral relocation is a prerequisite. Indeed, several arguments further support the notion that the basal pool is the main site of Notch activation. First, in *numb* mutant cells, Notch accumulates basally during cytokinesis, likely contributing to the ectopic activation of Notch in the pIIb daughter cell (see below, [72,73,80]. Second, newly synthesized Delta is primarily detected along the basal pIIa-pIIb interface, where it resides for a short period of time. Preventing Delta endocytosis by overexpressing an inhibitor of Neuralized activity leads to the accumulation of Delta mainly at the basal interface [73]. This indicates that Neuralized-mediated endocytosis of Delta, and hence activation of Notch in the pIIa cells, takes place from the basolateral membrane. Third, upon the blockade of g-secretase cleavage in *Presenilin* mutant cells, the unprocessed Notch localizes at the basal interface [73]. Having defined the site of Notch activation, in the next two paragraphs we will describe how cell fate determinants ensure the directionality of Notch signaling.

### 5.2. Mechanism of Notch Inhibition by Numb in the pIIb Cell 

Numb is a ubiquitous conserved protein containing a phospho-tyrosine binding domain, localizing in endosomes and uniformly at the cell cortex of interphase epithelial cells [19,61,81,82]. Numb was the first cell fate determinant related to the Notch pathway to be identified [12]. In *numb* mutants, the SOP divides symmetrically into two pIIa cells, while overexpression of Numb in the SOP leads to the opposite cell fate transformation, e.g., two pIIb cells, indicating that Numb represses Notch activation. In a mitotic SOP, Numb is located asymmetrically in the antero-basal cortex of the cell and is inherited by the pIIb cell [12]. Numb binds to and polarizes the distribution of a-Adaptin (a-Ada)—one of the four subunits of the AP-2 complex involved in receptor-mediated endocytosis—at the anterior cortex of dividing SOPs [83] (Figure 3A). The fact that mutant forms of a-Ada unable to bind to Numb cause *numb*-like phenotype and that Numb was reported to bind to the intracellular domain of Notch led to a model stating that Numb prevents Notch activation in the pIIb cell by mediating Notch endocytosis, i.e., the removal of Notch from the plasma membrane [17,83,84]. The identification of *spdo*, a gene that specifically regulates Notch-dependent asymmetric divisions, called into question the direct interaction between Numb and Notch [85,86]. Indeed, Spdo encodes a four-pass transmembrane protein that physically interacts with Numb, with the intracellular domain of Notch as well as with Presenilin, one subunit of the g-secretase complex, the Notch cleaving enzyme [87]. Spdo localizes at the plasma membrane in the pIIa cell where it facilitates the g-secretase cleavage of Notch and is required to specify the Notch-dependent fate of the pIIa cell. In contrast, in the pIIb cell, Numb prevents Notch signaling by inhibiting Spdo-Notch plasma membrane localization [18,19,85,86,87,88,89]. In addition, independently of its role in Numb localization, Lgl also inhibits the plasma membrane localization of Spdo in the pIIb cell [62]. The live-imaging of GFP-labelled Numb and Spdo revealed that during SOP cytokinesis Numb is relocated from the basal cortex to apically located endosomes in pIIb cells [19]. Endocytosis assays revealed that Spdo is internalized and accumulates with Numb in these endosomes. Importantly, GFP-tagged Notch also distributes in these apical sorting endosomes at cytokinesis, in a Numb-dependent manner [19]. Although Numb-AP-2 interaction is required for the cell fate acquisition and AP-2 is required for Spdo endocytosis, Spdo and Notch endocytosis are independent of Numb [18,19]. These data demonstrated that Numb does not regulate the distribution of Spdo and Notch via their endocytosis. Interestingly, Numb was shown to control post-endocytic trafficking and degradation of Notch in vertebrates and negatively regulate basolateral recycling in *C. elegans* [90,91]. Similarly, in *Drosophila* Numb genetically and physically interacts with the Clathrin adaptor AP-1 complex, a complex regulating the basolateral recycling of Spdo [75]. Accordingly, in a *numb* mutant sensory organ, Spdo is efficiently internalized and recycled back to the plasma membrane instead of being sorted towards late endosomes for degradation [18,19]. These data gave rise to a model according to which Numb binds to Notch-Spdo in apical endosomes and inhibits their recycling to the plasma membrane in an AP-1-dependent manner, and by so doing, Numb generates an asymmetry of Notch distribution along the pIIa-pIIb interface (Figure 3B). 

Since Numb does not regulate Spdo-Notch endocytosis, the requirement of Numb-AP2 interaction for Notch inhibition raises the question of the function of Numb-AP2 binding. The fact that Numb relocation to apical sorting endosomes is compromised upon loss of Numb suggests that AP-2 promotes the redistribution of Numb from the basal cortex to apical endosomes [19]. Furthermore, it cannot be ruled out that Numb-mediated AP-2 asymmetry generates an asymmetry in endocytosis capacity in the pIIb cell that is necessary for the endocytosis of components of the Notch pathway other than Spdo-Notch, and necessary for the acquisition of binary identities.

### 5.3. Mechanisms of Action of Neuralized in the pIIb Cell 

Neur was the second cell fate determinant identified in the SOP that, similarly to Numb, localizes at the antero-basal cortex of the cell in mitosis, and partitions unequally in the pIIb cell at cytokinesis [92]. In pIIb cell, Neur ubiquitinates the Notch ligand Delta to trigger its endocytosis and promotes Notch activation in the pIIa cell [14,15,16,92,93]. Neur-mediated endocytosis of Delta is proposed to lead to the activation of Notch in trans in the pIIa cell. This is because Notch activation relies on mechano-transduction [39,94]. Indeed, Notch requires a conformational change in the negative regulatory region of its extracellular domain (NRR domain), which masks the S2 cleavage site [95,96]. Experiments performed ex vivo in vertebrate cells revealed that a conformational change in the Notch NRR domain can be mechanically induced by pulling forces in the range of 4–9 pN [97,98,99,100]. In the quest of identifying the force-generator-mediating receptor activation, Epsin- and Clathrin-dependent endocytosis of the Notch ligand Delta was shown to generate pulling forces sufficient enough to activate Notch [101,102]. However, Epsin, which is generally required for ligand endocytosis and Notch activation in flies, is dispensable for SOP intra-lineage decisions [103]. Instead of Epsin, WASp and Arp2/3, known to increase endocytosis efficiency through the nucleation of branched actin [104,105], are specifically required for binary fate decision in the SOP lineage [106,107]. Careful analyses by time lapse revealed that in the pIIb cell, WASp, by activating Arp2/3, is required for Delta endocytosis at the time of Notch activation during cytokinesis [74]. The current model is that during cytokinesis, unequally inherited Neuralized ubiquitinates Delta in the pIIb cell, serving as a signal to trigger Delta endocytosis. Upon endocytic vesicle formation, WASp-Arp2/3 would provide an extra pushing force required for efficient endocytosis of the Delta bound to the Notch receptor in trans in the pIIa cell, in turn inducing the potent pulling forces on the Notch receptor present in the pIIa cell, hence Notch activation (Figure 4). The requirement for WASp-Arp2/3 activity instead of Epsin would be that intra-lineage decisions take place during cytokinesis at a time when Epsin activity could be negatively regulated by phosphorylation [74,108] and membrane rigidity would be higher, requiring more forces to deform it. Along this line, plasma membrane stiffness in the pIIa cell, the Notch-receiving cell in cytokinesis, may also play a role in signaling efficiency as cells with a rigid membrane are expected to experience more pulling forces from ligand endocytosis in the sending cell [109]. 

Although the role of Neur and endocytosis in binary fate decisions is now established, there are still some open questions: (i) While Neur behaves as a cell determinant, whether its asymmetric localization is strictly required for pIIa-pIIb fate acquisition remains to be determined. First, the *neur* mutant phenotype is rescued by expressing Mindbomb1, an ubiquitin ligase required for Notch activation outside of intra-lineage decisions, which is not asymmetrically localized [110]. Second, in *numb* mutant cells, while Neur is unequally inherited by the anterior cell, Delta from the posterior cell is able to activate Notch in the anterior cell that adopts the pIIa-like fate [92]. (ii) It has recently been proposed that Neur and Delta form a signaling complex in which Neur ubiquitinylates components other than Delta to enhance the signaling [111]. (iii) Neur exerts a Notch-independent function, via Stardust-dependent endocytosis of Crumbs, to regulate the apical surface of epithelial cells during epithelial morphogenesis [112,113,114]. It remains to be determined whether Neur activity is responsible for the localization of Crumbs in the apical endosomes of SOP daughter cells [55] and if this has an impact on the apico-basal polarity of the SOP and on the signaling at pIIa-pIIb interface, hence linking cell polarity remodeling and Notch signaling in this context.

### 5.4. Additional Contribution of Membrane Traffic in Binary Cell Fate Acquisition 

Over the last two decades, several events and/or regulators of membrane trafficking acting in addition to Numb and Neur have been identified and are likely to contribute to ensuring robustness and directionality in intra-lineage decision making.

As stressed in the previous paragraph, endocytosis plays an essential role in binary cell fate decision. Studies from the Gonzalez-Gaitan laboratory identified a population of Delta and Notch that are internalized in the SOP prior to mitosis, targeted to endosomes marked by the protein Sara (Smad anchor for receptor activation) [115]. During early cytokinesis, the Sara-positive endosomes (Sara endosomes) are transported to the central spindle thanks to the microtubule motor Klp98A. As the central spindle is made up of antiparallel microtubules, Sara endosomes are transported bidirectionally, until Klp10- and Patronin-mediated symmetry breaking of the central spindle takes place, leading to the unequal inheritance of Sara endosomes by the pIIa cell [116]. Notch itself, by binding to Uninflatable, a transmembrane protein containing EGF repeats, was shown to regulate the targeting to Sara endosomes and their asymmetric partitioning [117]. In Sara endosomes inherited by the pIIa cell, Notch was reported to undergo g-secretase cleavage [115]. Collectively, these data led to a model in which unequal segregation of Sara endosomes increases Notch signaling in the pIIa cell while decreasing it in the pIIb cell [115,116]. Although the characterization of the molecular mechanisms underlying the asymmetric segregation of Sara endosomes containing Delta and Notch is remarkable, their precise contribution to Notch activation remains debated [9,118]. This is because preventing the asymmetric distribution of Sara endosomes does not lead to a defect in Notch activation in the SOP lineage, excepted in the case where a dominant negative version of Rab5 is over-expressed [115], or in the case of depletion of Neuralized [116]. However, these situations do not allow the effect of the loss of Sara endosome asymmetry to be disentangled from the effect of Rab5 on the endocytosis required for the activities of Numb and Neuralized on the one hand, and the role of Neuralized as a cell fate determinant in regulating Delta endocytosis during cytokinesis on the other. Finally, the loss of function of Uninflatable, which causes a penetrant defect in the distribution of Sara endosomes, does not lead to a defect in pIIa/pIIb decision, but only to a weakly penetrant defect in the pIIa and pIIb daughter cell lineages [115]. At last, the tracking of functional GFP and a mCherry-tagged version of Notch revealed that the bulk of Notch-containing endosomes are equally partitioned at SOP cytokinesis [118]. Although this study does exclude the possibility that a subset of specialized endosomes containing Notch traffic directionally into pIIa at cytokinesis, collectively these data indicate that further, more discriminating studies should be set up to firmly establish their role in the directionality of Notch signaling.

Among the additional membrane traffic regulators are the Rab11 recycling endosomes, whose distribution differs between SOP daughters [119]. Indeed, Rab11 recycling endosomes in which Delta is trafficked are grouped together at the centrosome in the anterior pIIb cell, leading to the notion that Delta is recycled more efficiently and rapidly in pIIb, thereby increasing Delta signaling in the pIIb cell [119]. This notion was supported by the discovery that Sec15, a Rab11-interacting subunit of the exocyst complex and Epsin15 homology domain-containing protein-binding protein 1 (dEHBP1), regulates Delta recycling and is required for Notch activation in the pIIa cell [120,121]. However, it remains to be determined whether this recycling of Delta is required for Notch activation. Why would recycling be necessary? It is interesting to note that Delta-GFP is barely detected at the plasma membrane unless its endocytosis is prevented, suggesting that its time residence at the plasma membrane is short [73]. Thus, Rab11-Sec15-dependent recycling might be a means to constantly supply enough Delta to the signaling pIIa-pIIb interface. An alternative would be that the recycling enables the targeting of Delta to the proper plasma membrane domain. The Arp2/3 and WASp complex, in addition to being required for boosting Delta endocytosis at cytokinesis [74], is required for the formation of a specific actin-rich apical structure at the SOP daughter interface and for the trafficking of Delta into this structure [107]. Because the photo-tracking of a fluorescently tagged Notch receptor shows that Notch activation takes place primarily at the lateral membrane, Rab11-Sec15-dependent recycling of Delta may serve to relocate Delta from its apical delivery site to the lateral plasma membrane. Further work would help in deciding between these possibilities. In addition, since Sec15, Rab11 and Arp2/3-WASp also regulate the trafficking of other transmembrane proteins that impact cellular mechanics and polarity [122,123,124], it may be possible to envisage that other targets than Delta contribute to Notch-dependent binary fate acquisition.

## 6. Actual Limitations and Concluding Remarks 

The genetic and live-imaging dissection of the seemingly simple SOP lineage has allowed the decryption of the molecular mechanisms underlying the regulation of fate asymmetry following asymmetric division. The development of microscopes with more sensitive detectors and improved axial resolution, enabling better spatio-temporal photo manipulation and photo-tracking, will certainly help better monitoring Notch receptor trafficking, proteolytic activation and translocation in the nuclei in the 3D context of epithelial cells. Approaches such as retention using selective hooks (RUSH) [125] would enable a real-time visualization of the exocytosis of ligands and receptors to pinpoint where they are addressed in polarized cells. The spatio-temporal control of the activity of Notch regulators, thanks to the use of thermosensitive mutants, knockout-side or optogenetic, would be informative to decipher the time and location of Notch activation (prior, during of following SOP division). 

As stressed in this review, some questions are still open or debated. Among them is the role of epithelial junctional remodeling in asymmetric division, which has begun to emerge. The mechanism by which NICD is translocated in the nuclei following proteolytic activation at the plasma membrane is also an important question. Notch activation is initiated in the pIIa cell at a time where the two daughter cells are still connected by the midbody through which fluorescent probes of similar size to NICD are equilibrated between pIIa and pIIb within a few minutes [126]. How is directionality to restrict nuclear import to the pIIa cell achieved? Based on the fact that SOP division is a semi-closed mitosis (C. Roubinet and RLB, unpublished observations) and that importins regulate Notch signaling [127,128], it will be interesting to test if an asymmetry in nuclear envelope and/or nuclear pore reformation between pIIa and pIIb could also contribute to binary fate acquisition. 

Finally, as vertebrate Notch is involved in the regulation of stem cell self-renewal and differentiation [2], in solid tumors [3], it can be anticipated that the findings obtained in the *Drosophila* SOP model system will continue to be useful to decipher the molecular and cellular basis of the activation of this fundamental paracrine communication pathway, and ultimately propose ways to modulate it for curative purposes. 

## Figures and Tables

**Figure 1 cells-13-01133-f001:**
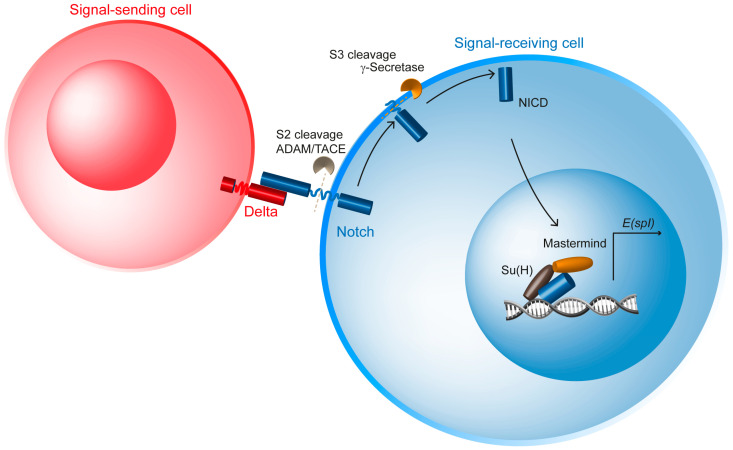
Overview of the Notch pathway in *Drosophila.* The Notch pathway is a paracrine communication pathway involving Delta (red), the transmembrane ligand expressed and localized at the plasma membrane of the so-called signal-emitting cell, depicted in red. Delta interacts in trans with the Notch receptor (blue), localized at the plasma membrane of the signal-receiving cell, depicted in blue. Following ligand binding, Notch undergoes two successive proteolytic cleavages. The first cleavage at S2 is ligand-induced and generates an activated membrane form of Notch, which is then processed at S3 by the γ-secretase complex. This leads to the release of NICD, its translocation into the nucleus, and association with the transcriptional co-activator Mastermind and the DNA-binding transcription factor Suppressor of Hairless (Su(H)), resulting in transcriptional activation.

**Figure 2 cells-13-01133-f002:**
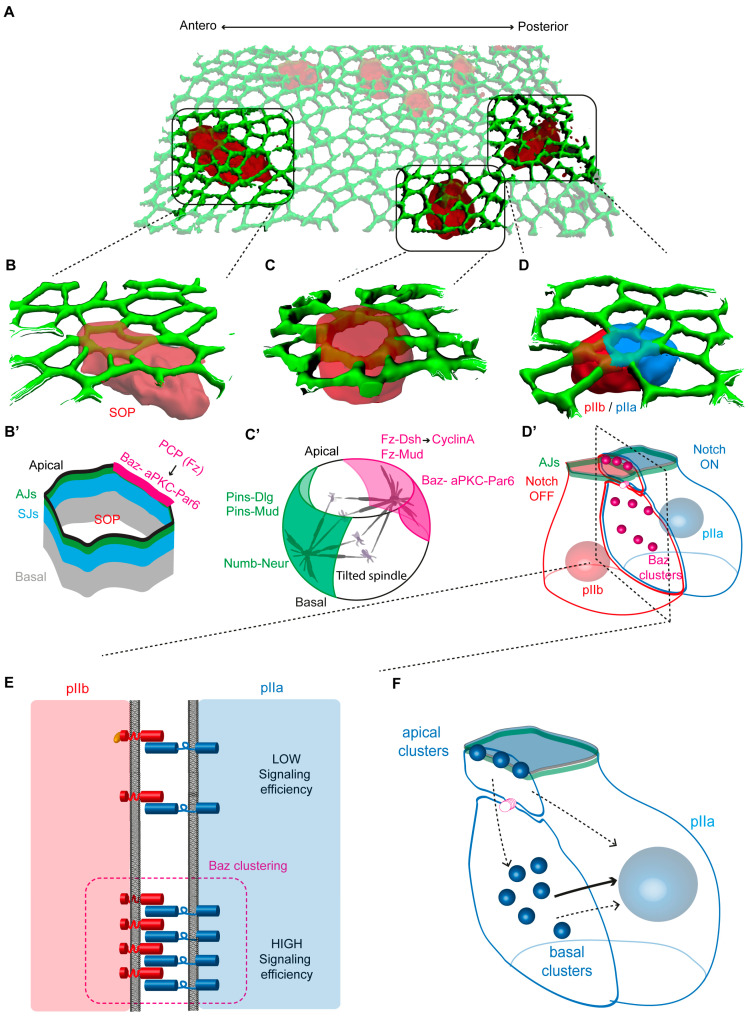
Asymmetric division of the SOP and site of Notch activation in cytokinesis. (**A**) Three-dimensional view of the dorsal thorax pupal notum, a single-layer epithelium composed of epidermal cells and SOPs (red cells). Adherens junctions labelled by E-cadherin (green) are localized apically and delimit the contour of epithelial cells. SOPs are regularly spaced thanks to Notch-dependent lateral inhibition. (**B**–**D**) High magnifications of the SOP in interphase (**B**), during mitosis (**C**) and at mitosis exit once the SOP daughters, the anterior pIIb (red) and posterior pIIa (blue) cells, are formed (**D**). (**B**′) Schematic representation of the SOP in interphase illustrating the position of cell–cell junctions, adherens junctions (AJs, green) and septate junctions SJs (blue). PCP-dependent planar asymmetry of Baz-aPKC-Par6 (magenta) is initiated prior to mitosis. (**C**′) Schematic representation of the SOP in prometaphase. The PCP complex (Fz-Dsh) localized at the posterior apical cortex, together with the Baz-aPKC-Par6 polarity complex, (magenta) while Pins-Dlg co-localize with Numb and Neur at the antero-basal cortex (green). Mud interacts with Pins at the anterior basal cortex, and with Fz-Dsh at the antero-posterior cortex. In turn, Mud, by interacting with the Dynein complex, causes the spindle to orient along the anterior–posterior axis with a slight apical–basal tilt of the spindle. (**D**′) schematic representation of the pIIa-pIIb SOP daughter cells. The apical–basal polarity of the SOP daughter is preserved despite the polarity remodeling that has taken place in mitosis. However, specific to pIIa-pIIb cells (not seen following cytokinesis of epidermal cells) is the assembly of Baz clusters at the apical and the lateral interface. (**E**) Baz clusters contain Notch, Delta and Spdo, and Baz activity is required for efficient Notch signaling. (**F**) Photo-tracking of NICD revealed that the main contributors in Notch signaling originates from the basal pool of Notch (arrow). While the apical pool also contributes to Notch signaling, it remains to be determined if proteolytic activation occurs directly at the apical plasma membrane or if it is relocated from the apical to basal plasma membrane (dotted arrow).

**Figure 3 cells-13-01133-f003:**
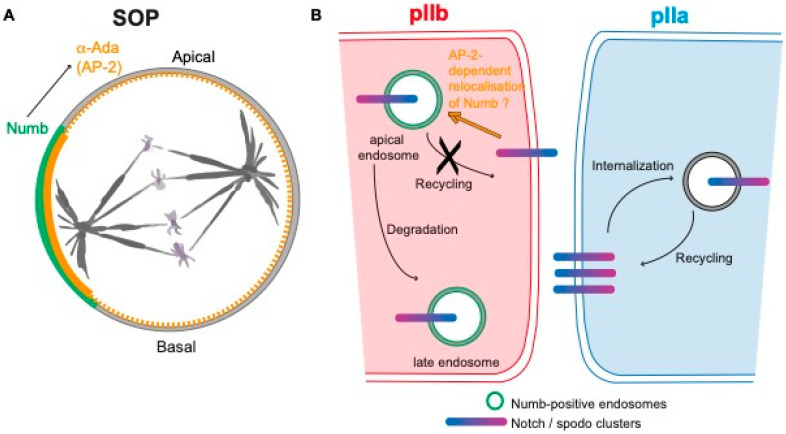
Numb regulates the recycling of Notch-Sanpodo in the pIIb cell. (**A**) During SOP division, Numb (green) localizes at the anterior cortex and promotes the asymmetric distribution of a-Adaptin (α-Ada, orange), one of the subunits of the AP-2 complex, regulating Clathrin-mediated endocytosis. (**B**) During cytokinesis, Numb is unequally partitioned in the pIIb cell, where it is relocated from the cortex towards apical endosomes in an a-Ada-dependent manner. There, Numb prevents the recycling of Notch-Spdo (blue/purple) to the plasma membrane in favor of late endosomal transport. In contrast, in the pIIa cell, in the absence of Numb, Notch-Spdo can be recycled to the plasma membrane. According to this model, where Numb negatively regulates the recycling of Notch-Spdo, Numb, by reducing the level of Notch present at the plasma membrane of the pIIb cell, generates the asymmetrical low Notch/high Notch levels in pIIb and pIIa cells, respectively.

**Figure 4 cells-13-01133-f004:**
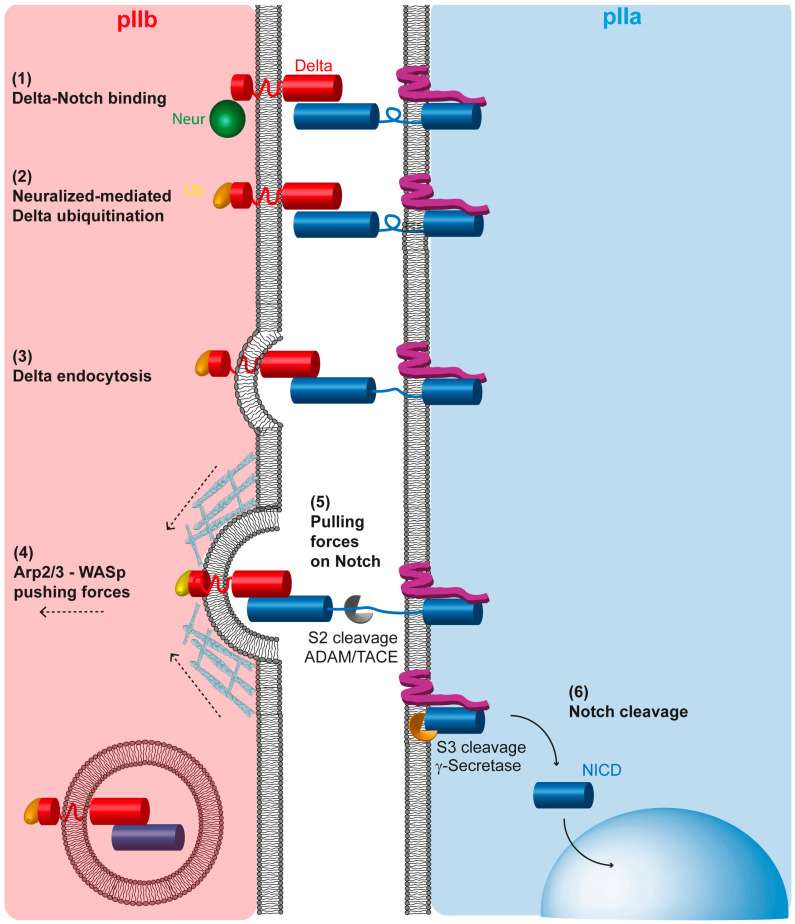
Neur regulates Delta endocytosis in the pIIb cell to trigger mechanosensitive activation of Notch in the pIIa cell. Activation of Notch (blue in the pIIa cell) is a multi-step process. It requires first the binding of the ligand Delta (red in pIIb) in trans (1). Then, Neur (green) ubiquitinates Delta (post-translational modification of the cytoplasmic domain of Delta depicted in yellow) (2). The ubiquitin moiety serves as a signal triggering Delta internalization (3). Arp2/3- and WASp-dependent polymerization of branched actin promotes extra pushing forces on the forming endocytic vesicle (4). Delta is bound to Notch, and Delta endocytosis exerts pulling forces on Notch, (5) thereby causing a conformational change in the Notch extracellular domain which reveals the S2 cleavage site for ADAM secretase. The second proteolytic cleavage triggered by g-secretase (cleavage S3) occurs, releasing into the cytoplasm of the pIIa cell the Notch intracellular domain, which can then be translocated into the nucleus.

**Table 1 cells-13-01133-t001:** **Core components of the Notch signaling pathway**, adapted from [39].

Function	Type	Invertebrates (*Drosophila*)	Vertebrates (Mammals)
**Receptor**		Notch	Notch 1, 2, 3 and 4
**Ligand**		Delta Serrate	Dll1, 3 and 4 Jagged1 and 2
**Nuclear effectors**	CSL DNA-binding transcription factor Transcriptional coactivator	Su(H) Mastermind	RBPjk/CBF-1 MAML1-3
**Receptor proteolysis**	Site 1 cleavage Site 2 cleavage Site 3 cleavage	Kuzbanian Kuzbanian-like; TACE Presenlin, Nicastrin APH-1, PEN-2	PC5/6, Furin ADAM10/Kuzbanian ADAM17/TACE Presenlin 1 and 2, Nicastrin APH-1a-c, PEN-2
**Membrane trafficking**	E3 Ubiquitin ligase Negative regulators Neuralized inhibitors others	Mindbomb 1 and 2 Neuralized Numb Bearbed, Tom, M4 Sanpodo	Mindbomb, Skeletrophin Neuralized 1 and 2 Numb, Numb-like, ACBD3
**Canonical target bHLH repressor genes**		*E(spl)*	*HES/ESR/HEY*

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
