# Peer review of "Spatio-Temporal Regulation of Notch Activation in Asymmetrically Dividing Sensory Organ Precursor Cells in Drosophila melanogaster Epithelium"

_cells, 2024, doi:10.3390/cells13131133_

Round 1

Reviewer 1 Report

Comments and Suggestions for Authors

In this review, the authors beautifully summarized the mechanisms underlying asymmetric cell division of sensory organ precursor (SOP) cells. The authors are clearly experts in the field of Notch signaling and they presented a textbook level review showing the current knowledge about SOP polarization, the location of Notch signaling, and the mechanism of differentiated regulation of Notch in pIIa and pIIb daughter cells. I have no major comments with this elegant manuscript except a few minor considerations. Overall, I strongly recommend the publication of this paper on cells.

Minor point:

1.       Notch signaling is conserved in vertebrates (as the authors suggested). It would be beneficial to include a small paragraph to discuss the vertebrate homologs of Notch pathway components.

2.       Line 23. Missing right parentheses.

3.       Line 55. I am assuming the authors are referring to asymmetric cell division, but the abbreviation “ACD” is not defined until Line 128.

4.       Line 426. Typo “… by g-secretase (cleavage S3)”, should be γ secretase.

Author Response

Answer to Reviewer 1:

In this review, the authors beautifully summarized the mechanisms underlying asymmetric cell division of sensory organ precursor (SOP) cells. The authors are clearly experts in the field of Notch signaling and they presented a textbook level review showing the current knowledge about SOP polarization, the location of Notch signaling, and the mechanism of differentiated regulation of Notch in pIIa and pIIb daughter cells. I have no major comments with this elegant manuscript except a few minor considerations. Overall, I strongly recommend the publication of this paper on cells.

We would like to thank reviewer 1 for his critical comments and his recommendation to publish the review.

Minor point:

  1. Notch signaling is conserved in vertebrates (as the authors suggested). It would be beneficial to include a small paragraph to discuss the vertebrate homologs of Notch pathway components.

In the revised version, we have included Table 1, which lists the main components of the Notch pathway in mammals, and added the corresponding text on lines 107-108.

  1. Line 23. Missing right parentheses.

Parenthese added

  1. Line 55. I am assuming the authors are referring to asymmetric cell division, but the abbreviation “ACD” is not defined until Line 128.

The reviewer is right, the abbreviation has been defined (lines 54-55).

  1. Line 426. Typo “… by g-secretase (cleavage S3)”, should be γ secretase.

corrected

Reviewer 2 Report

Comments and Suggestions for Authors

The Notch signaling was first discovered in Drosophila to regulate asymmetric cell division of sensory organ precursors. It is a classic regulatory pathway and a lot of reviews have summarized the Notch function during various developmental processes. In this review, the authors focus on how epithelial cell polarity, cell cycle and intracellular trafficking participate in controlling the directionality, subcellular localization and temporality of Notch receptor to decipher the complexity of Notch signaling in asymmetric cell division.

Here are some comments and questions that need to be addressed.

1. There are similar reviews that have been published last year, such as Rodriguez et al., 2023 Current Opinion in Cell Biology, 85:102244, and the authors did not cite this essential review. Additionally, a certain amount of important work of Notch signaling in asymmetric cell division in mammalian and cell lines has been published in the recent couple of years, however, there are almost no publications in Drosophila. I doubt the innovation and necessity of this manuscript.

2. There are 12 subtitles in this review, which looks not focused. Actually, the authors stated in lines 55-62 that they would like to discuss three questions after introducing the basic knowledge of Notch signaling. A concise subtitle corresponding to each question is recommended in the following text, but not several separated subtitles as shown from 3 to 11 in the present version.

3. In Figure 1, the authors stated that Notch activation relies on the sequential cleavage of S2 and S3. The transmembrane domain of Notch is clearly shown in Figure 1, but the cleavage sites of S2 and S3 are not reflected due to the design of ADAM/TACE and γ-secretase.

Author Response

The Notch signaling was first discovered in Drosophila to regulate asymmetric cell division of sensory organ precursors. It is a classic regulatory pathway and a lot of reviews have summarized the Notch function during various developmental processes. In this review, the authors focus on how epithelial cell polarity, cell cycle and intracellular trafficking participate in controlling the directionality, subcellular localization and temporality of Notch receptor to decipher the complexity of Notch signaling in asymmetric cell division.

We would like to thank the reviewer for her/his time and critical comments.

Here are some comments and questions that need to be addressed.

There are similar reviews that have been published last year, such as Rodriguez et al., 2023 Current Opinion in Cell Biology, 85:102244, and the authors did not cite this essential review.

We are grateful to the reviewer for bringing to our attention this remarkable review from the laboratory of C. Salhgren which had escaped us. We have now included it on line 400, next to that of the laboratory of R. Kopan and Ilagan.

  1.  

Additionally, a certain amount of important work of Notch signaling in asymmetric cell division in mammalian and cell lines has been published in the recent couple of years, however, there are almost no publications in Drosophila. I doubt the innovation and necessity of this manuscript.

Our manuscript is an invited review on the contribution of the Drosophila model to the understanding of Notch signalling during asymmetric division of SOPs. We leave it to the editors to decide on the innovation and necessity of this manuscript.

  1. There are 12 subtitles in this review, which looks not focused. Actually, the authors stated in lines 55-62 that they would like to discuss three questions after introducing the basic knowledge of Notch signaling. A concise subtitle corresponding to each question is recommended in the following text, but not several separated subtitles as shown from 3 to 11 in the present version.

We have taken this criticism into account. After the introductory part, we have modified the titles and subtitles (text in green in the revised version) to answer three main questions:

1- Establishment of the asymmetry of cell fate determinants

2-Remodeling of cell-cell contacts during SOP cytokinesis

3- Role of polarized trafficking and endocytosis in Notch activation

and finish on:

Actual limitations and concluding remarks

We hope that this organisation will meet the Reviewer's expectations.

  1. In Figure 1, the authors stated that Notch activation relies on the sequential cleavage of S2 and S3. The transmembrane domain of Notch is clearly shown in Figure 1, but the cleavage sites of S2 and S3 are not reflected due to the design of ADAM/TACE and γ-secretase.

We took this comment into account and revised Figure 1 to illustrate the S2 and S3 cleavage sites.

Reviewer 3 Report

Comments and Suggestions for Authors

This manuscript by Pinot and Le Borgne is an up-to-date review on the regulation of Notch signaling during asymmetric cell division (ACD) in Drosophila sensory organ precursor cells (SOPs) model system. This is an interesting topic as, beside the fundamental aspect, improvement of the understanding of the control mechanisms of Notch signaling will certainly lead to the development of new therapeutic approaches in various diseases, such as cancer, where Notch signaling is altered. The review is well written and logically organized. It is a long manuscript, which contains a substantial amount of information and references. The bibliography analysis is sound and rather exhaustive. In my opinion, this paper is worth to be published in Cells.

Author Response

This manuscript by Pinot and Le Borgne is an up-to-date review on the regulation of Notch signaling during asymmetric cell division (ACD) in Drosophila sensory organ precursor cells (SOPs) model system. This is an interesting topic as, beside the fundamental aspect, improvement of the understanding of the control mechanisms of Notch signaling will certainly lead to the development of new therapeutic approaches in various diseases, such as cancer, where Notch signaling is altered. The review is well written and logically organized. It is a long manuscript, which contains a substantial amount of information and references. The bibliography analysis is sound and rather exhaustive. In my opinion, this paper is worth to be published in Cells.

We would like to thank the reviewer for her/his positive comments and for supporting the publication of the Review.

Round 2

Reviewer 2 Report

Comments and Suggestions for Authors

The authors have responded my questions and revised the manuscript. I have no further questions.